# A Novel Strategy to Reveal the Landscape of Crossovers in an F1 Hybrid Population of *Populus deltoides* and *Populus simonii*

**DOI:** 10.3390/plants11081046

**Published:** 2022-04-12

**Authors:** Zhiting Li, Wei Zhao, Jinpeng Zhang, Zhiliang Pan, Shengjun Bai, Chunfa Tong

**Affiliations:** Co-Innovation Center for Sustainable Forestry in Southern China, College of Forestry, Nanjing Forestry University, Nanjing 210037, China; lizhiting@njfu.edu.cn (Z.L.); zhaowei@njfu.edu.cn (W.Z.); zhangjinpeng@njfu.edu.cn (J.Z.); panzhiliang@njfu.edu.cn (Z.P.); sjbai@njfu.edu.cn (S.B.)

**Keywords:** crossover rate, crossover interference, *Populus deltoides*, *Populus simonii*, F1 hybrid population

## Abstract

Although the crossover (CO) patterns of different species have been extensively investigated, little is known about the landscape of CO patterns in *Populus* because of its high heterozygosity and long-time generation. A novel strategy was proposed to reveal the difference of CO rate and interference between *Populus deltoides* and *Populus simonii* using their F1 hybrid population. We chose restriction site-associated DNA (RAD) tags that contained two SNPs, one only receiving the CO information from the female *P. deltoides* and the other from the male *P. simonii*. These RAD tags allowed us to investigate the CO patterns between the two outbred species, instead of using the traditional backcross populations in inbred lines. We found that the CO rate in *P. deltoides* was generally greater than that in *P. simonii*, and that the CO interference was a common phenomenon across the two genomes. The COs landscape of the different *Populus* species facilitates not only to understand the evolutionary mechanism for adaptability but also to rebuild the statistical model for precisely constructing genetic linkage maps that are critical in genome assembly in *Populus*. Additionally, the novel strategy could be applied in other outbred species for investigating the CO patterns.

## 1. Introduction

Genetic recombination is a fundamental biological process during meiosis. It not only affects the adaptability of organisms to the environment [1], but also plays an important role in the biological evolution and formation of new species [2]. Because recombination rate is approximately proportional to the physical distance between molecular markers, it also provides the basis for genetic mapping [3]. Recombination is generated by double strand breaks (DSBs) in a chromosome [4]. With the repair of DSBs through the double Holliday junction or synthesis-dependent strand annealing pathways, two results are possibly produced, i.e., crossovers (COs) and non-crossovers (NCOs) [5,6]. COs are the reciprocal exchanges of DNA fragments between homologues, whereas NCOs are the recombination products where a section of genetic information is copied from one homologous chromosome to the other, without affecting the donating chromosome. Consequently, a CO affects two chromatids, but an NCO affects only one chromatid [5,6]. In most species, the number of COs occupies a small fraction of the DSBs [7,8]. However, because of the creation of physical connection between homologous chromosome pairs, COs are essential for faithful chromosome segregation in meiosis [9]. Moreover, because NCOs are very short and difficult to detect [10,11], most studies used the rate of CO events to represent the recombination rate [12,13,14].

Many studies have shown that the CO rates are quite dissimilar in different organisms, and the distribution of CO events varies greatly even in closely related taxa, populations and individuals [15,16]. Among them, the most striking is that the CO rate of microorganisms and fungi is much higher than that of animals and plants [17]. In fact, this comparison is more common in the specific taxonomic groups. For example, among mammals, marsupials have lower CO rates [18]; and among plants, conifers have very low CO rates [19]. Moreover, a large number of articles have proved that the CO rates of different organisms are related to the genome architecture and chromatin structure [20,21,22,23]. In addition, selection pressure and environmental factors may also lead to the difference of CO rates [24,25].

COs at both chromosomal and local scales are strictly controlled by a pervasive mechanism [26,27], and they do not arise as independent events. The mechanism is called CO interference [28], by which the occurrence of a CO affects the formation of other COs in its vicinity. Early studies of genetic maps showed that the existence of a CO in one chromosomal interval reduces the probability of another CO occurring nearby [29], a phenomenon called positive interference. However, subsequent studies in maize [30] and *Arabidopsis thaliana* [12] found the opposite phenomenon, that is, CO interference could increase the probability of occurrence of nearby COs (negative interference). CO interference is extremely different in some species. For example, in the meiosis of *Caenorhabditis elegans*, each bivalent produced exactly one CO, which indicated that there existed complete interference [31], whereas there was weak or no interference in *Aspergillus nidulans* [32]. Apart from these extreme cases, CO interference also varies greatly in different species. For example, the strength of interference across the dog genome was estimated to be much higher than that in the human genome [33]. Another noteworthy phenomenon is that there is compelling evidence that females and males also have disparate interference landscapes [33,34]. In *Anser anser* [35], *Canis familiaris* [33], *Homo sapiens* [36,37], and *Prunus mume* [28], the CO interference in females is lower than that in males. In contrast, in *Bos Taurus* [38,39] and *A. thaliana* [12,40], the CO interference is higher in females than in males. Some studies indicated that the difference of CO interference between the sexes may be the main factor leading to the sexual difference in CO rates [34].

The genus *Populus* was used as a model system in forest genetics and woody plant studies [41]. Although numerous linkage maps have been constructed with different types of genetic markers in the genus [42,43,44,45,46], little is known about the CO patterns in different poplar species. Wang et al. [47] compared the average CO rate over 100-kbp nonoverlapping windows of three related poplar species, but there were no reports about the genome-wide difference in CO patterns between the species. In this study, we reported the CO patterns across the genome in an F1 hybrid population derived by crossing a female *P. deltoides* and a male *P. simonii*. The F1 population was established in our previous studies and a large number of individuals were successively sequenced with the restriction site-associated DNA sequencing (RADseq) technology [43,46]. Single nucleotide polymorphism (SNP) genotypes of individuals were obtained by mapping their paired-end (PE) reads to the genome sequence of *P. simonii* [48]. We chose those RAD tags that contained two SNPs, one heterozygous in the female and homozygous in the male and the other vice versa, to investigate the difference of CO patterns between the two species. The result would facilitate not only understanding of the CO role in the evolution of organisms and species, but also rebuild the statistical model for precisely constructing genetic linkage maps, which are critical in mapping quantitative trait loci (QTLs) and genome assembly [49]. Meanwhile, the novel strategy of investigating the difference of CO pattern between two parents using an F1 hybrid population could be applied in other outbred species.

## 2. Results

### 2.1. SNP Genotype Data

A total of 48,938 SNPs were obtained across the two parents and 257 individuals in the F1 hybrid population of *P. deltoides* and *P. simonii* by mapping their RADseq reads to the male parent genome sequences of *P. simonii* [50]. The average number of RADseq reads was 18,416,890 for the parents and the progeny (Appendix A). After a stringent filtering procedure with the NGS QC toolkit [51], an average amount of 4.61 Gb HQ data per individual remained for calling SNP genotypes. As expected from previous studies in the same population [46,52], the majority of SNPs segregated in the ratio of 1:1 (*p* > 0.01), with 29,466 SNPs segregating in the type of *ab* × *aa* and 18,863 in *aa* × *ab*. The SNP genotype for each individual at each SNP site satisfied the requirements that the read coverage depth of an allele was ≥3 for heterozygosity and ≥5 for homozygosity, and that the Phred score was at least 30. Although the missing genotype rate was allowed to up to 20% at an SNP site, the average number of genotyped individuals was up to 239 per SNP site.

Based on the two SNP datasets of different segregation types, a total of 1157 RAD tags were selected for performing the comparison analysis of CO rate and interference between the female and male parents. These RAD tags spanned 95.34% of the reference genome. Each RAD tag contained two SNPs fulfilling the condition that one segregates in *ab* × *aa* and the other in *aa* × *ab* (Figure 1A), and that the distance between them is less than 1 Kb. Moreover, the distance between any two adjacent RAD tags was greater than 100 Kb so that the crossover events in the interval can be detected in the population. In addition, the distributions of the RAD tag number and interval length within chromosomes were outlined in Table 1. It can be seen that the average interval length was around 300 Kb in all chromosomes except chromosome 19, in which the average length was doubled up to 636 Kb.

### 2.2. Difference of CO Rates between Two Parents

We calculated the CO rates in each RAD tag interval using the SNPs with segregation types of *ab* × *aa* for the female parent *P. deltoides* and *aa* × *ab* for the male parent *P. simonii*. The range and mean of CO rates within chromosomes were presented for *P. deltoides* and *P. simonii* in Table 1. It can be observed that the minimum CO rate was 0.004 and the maximum up to 0.224 over the intervals on all chromosomes. The means of CO rates on all chromosomes but three (Chr08, Chr12, and Chr14) were greater in *P. deltoides* than in *P. simonii*. The maximum mean difference was found to occur in chromosome 5 and 11. The Wilcoxon signed-rank test showed that the difference of the mean CO rate between the two species was significant with a *p*-value of 0.002.

We tested the difference of CO rates between the two species in each interval with the LR statistic. Consequently, of all the 1138 RAD tag intervals, there existed 307 intervals in which the CO rates between the two species were significantly different (Figure 2). The distribution of the number of these significant intervals within chromosomes was summarized in Table 2. We found that two or more significant intervals were successive and formed a region that we called a significant region. The number of the significant regions within each chromosome was also summarized in Table 2. Meanwhile, the distribution of the length of significant intervals and regions was investigated and also presented in Table 2 for each chromosome. It can be seen that the lengths of most significant intervals and regions were less than 1 Mb, but there were 27 significant intervals or regions distributed on 13 chromosomes whose lengths were greater than 1 Mb. Furthermore, the coverage of the significant intervals of each chromosome was calculated to be ranged from 6.91% (Chr18) to 33.65% (Chr04), resulting in a total coverage of 23.80% of the whole genome (Table 2). In addition, we found that the maximum significant region contained three significant intervals and spanned 2.36 Mb on Chr04 (Figure 2).

### 2.3. Investigation of CO Interference in Both Species

Since the degree of CO interference between two adjacent marker intervals depends on the interval lengths, different scales of the SNP intervals were considered to investigate the landscape of CO interference across the genomes in the female *P. deltoides* and male *P. simonii*. We selected those SNPs such that the minimum interval length (MIL) was required to be 0.5, 1.0, 2.0, 3.0, 4.0, and 5.0 Mb separately for calculating the coefficient of coincidence (CoC). When the MIL was equal to 0.5 Mb, there were 454 pairs of adjacent SNP intervals, of which 290 and 233 showed significant CO interference in *P. deltoides* and *P. simonii*, occupying 79.86% and 69.77% of their genomes, respectively (Table 3, Figure 3). As the MIL increased, the number of adjacent interval pairs with CO interference (NPCI) decreased, covering over half of the genomes for most cases. When the MIL reached 4.0 Mb or more, the interference-occurred regions in the male *P. simonii* abruptly became small. If the MIL was equal to 5.0 Mb, there existed nine pairs of interference-occurred intervals, covering 22.78% of the male *P. simonii* genome. Similarly, when the MIL reached 5.0 Mb, the NPCI in the female *P. deltoides* decreased abruptly from 290 to 14, with the genome coverage from 79.86% to 35.10% (Table 3). Overall, except the case when the MIL was equal to 3.0 Mb, the NPCI was slightly larger in the female than in the male, indicating that *P. deltoides* has a higher level of CO interference.

We found that the number of the common pairs of interference-occurred intervals in both species decreased as the MIL increased (Table 3). When the MIL was set as 0.5 Mb, there were 168 common pairs of interference-occurred intervals, spanning 52.61% of the genomes. However, when the MIL increased to 5.0 Mb, only four common pairs of interference-occurred intervals existed, covering 9.20% of the genomes.

The positive and negative CO interferences were also investigated across the cases of different MILs (Table 3). It was observed that when the MIL was equal to 0.5 Mb, the number of pairs of negative interference intervals (CoC > 1) was significantly greater than that of positive interference intervals (CoC < 1), especially for the female *P. deltoides* (214 versus 76). For each species, we observed that the number of pairs of positive interference intervals generally decreased to 0 as the MIL increased from 0.5 to 5.0 Mb, indicating that the positive interference vanished when the interval length was equal to or greater than 5.0 Mb. Meanwhile, the number of pairs of negative interference intervals decreased to the minimum numbers for each species as the MIL increased from 0.5 to 5.0 Mb.

We further compared the difference of CO interference between the female *P. deltoides* and male *P. simonii* by calculating interference strength parameters within chromosomes. Figure 4 showed the estimated values of interference strength across the 19 chromosomes for the female and male. It was observed that the parameters were consistently less than 1, and those for the female were generally less than for the male in all chromosomes except in chromosome 4. Moreover, the result from the *t* test showed that the average value (0.559) of the female parameters was significantly less than that (0.611) of the male (*p*-value = 0.012), indicating the negative interference strength in *P. deltoides* was greater than that in *P. simonii*.

## 3. Discussion

We took a novel strategy to successfully reveal the landscape of genome-wide CO events in an F1 hybrid population of *P. deltoides* and *P. simonii*. Such work used to be considered a tough task in forest trees. In inbred lines, reciprocal backcrosses with the same set of molecular markers were usually utilized to investigate the similarities and differences of male and female recombination in plants such as in *Arabidopsis* [12,54], *Brassica nigra* [55], and maize [56]. However, it is not so easy to establish the BC populations in forest trees because of their specific biological characteristics such as high heterozygosity and long generation times. Although the so-called ‘pseudo-testcross’ was often applied for linkage mapping in forest trees [57], there exists a barrier that the linkage phase was not fixed so that the traditional software for linkage analysis cannot be directly used [58,59]. Yet, in the current study, we obtained RAD tags in the F1 hybrid population with the RAD sequencing technology [60,61]. These RAD tags contained two types of SNPs—one that segregated in *ab* × *aa*, and the other in *aa* × *ab* (Figure 1)—constituting two datasets just like from two separate ‘pseudo-testcross’ populations and permitting the investigation of the CO patterns between two parents in the same genome regions. This strategy of combining the use of the F1 hybrid population and the RAD sequencing technology for studying COs of two parents could be applied to other forest trees or even other outbred species.

However, the analysis of such a strategy cannot be directly implemented with the traditional statistical methods applied in the populations derived from inbred lines. Since the linkage phase between two SNP markers was not fixed in the ‘pseudo-testcross’ population, we first inferred the linkage phase from coupling and repulsion cases using two-point linkage analysis before comparing the CO rates between the two parents at the same region. Meanwhile, in the analysis of CO interference, different linkage phase combinations of two adjacent marker intervals were also considered to establish the likelihood for estimating the CoC and performing the corresponding hypothesis test (Appendix A). Unlike in the previous studies [12,54,55,56], we proposed a statistical strategy for analyzing the CO patterns in the two parents, which was based on the maximum likelihood framework and simultaneously incorporated the complex linkage phases in the F1 hybrid population.

We therefore revealed the genome-wide CO patterns between the two parents in the F1 population, though the result may not be suitable for all species in *Populus*. First, the CO rate of the female *P. deltoides* is generally greater than that of the male *P. simonii* across the genomes, with 23.80% of the genome regions in which the two parental CO rates were significantly different. This result is opposite to many studies which showed that, in angiosperms, the CO rate in females was lower than in males [62]. Second, we revealed that the occurrence of CO interference pervaded both parental genomes at local scale when the MIL was limited to 0.5 Mb, covering ~80% of the female *P. deltoides* genome and ~70% of the male *P. simonii* genome, where the negative interference was more frequent than the positive interference (Table 3). Moreover, we found that there existed a significant difference of CO interference between the two parents in that the negative interference in the female was stronger than in the male. The phenomenon of extensive occurrences of CO interference was also observed by Wang et al. [63] in *P. euphratica*. The difference of interference strength between two sexes in *Populus* was similar to the results from studies in mice and cattle [34,39], and even in humans [64]. However, our result is limited to the female *P. deltoides* and the male *P. simonii*. Therefore, extensive studies should be performed in other intraspecific or interspecific crosses to further confirm whether such a result is a common phenomenon in *Populus*.

Besides helping understand its roles in the evolution of species, the assessment of CO interference has great implications in modelling multi-loci linkage analysis for precisely constructing genetic linkage maps. Up to date, the most popular genetic mapping software such as JoinMap [65] and MapMaker [66] did not incorporate the situation of CO interference either in two- and three-point linkage analysis or in multiple locus ordering. If CO interference significantly existed in meiosis, linkage maps constructed with these tools certainly had some limitations in the estimated genetic distances between markers and the marker orders within linkage groups, possibly seriously affecting the downstream analyses such as map-based QTL mapping [67,68] and genome sequence assembly [50,69]. Therefore, it is necessary to develop new algorithms by incorporating the CO inference parameters to calculate the genetic distances and the likelihood for an order of a large number of marker loci. This would substantially enhance the accuracy of genetic maps and thus facilitate the assembly of contigs or scaffolds into chromosomes, which is an important task in the era of high-throughput genomic sequencing [49]. It is worth noting that the CO interference affects QTL studies in two aspects. On the one hand, genetic linkage maps that incorporated CO interference could improve the power of detecting QTLs. On the other hand, the popular QTL statistical methods such as interval mapping [67] and composite interval mapping [68] were established under the assumption of without CO interference. Without any doubt, if CO interference was incorporated into these QTL mapping models, it would greatly improve their ability to identify QTLs.

## 4. Materials and Methods

### 4.1. Plant Material and Sequencing Data

A total of 257 individual trees were selected from the F1 hybrid population of *P. deltoides* and *P. simonii* for investigating the difference of CO patterns between the two parents. The F1 population was established by crossing the female *P. deltoides* and the male *P. simonii* from 2009 to 2011, as described in Tong et al. [46] and Mousavi et al. [43]. The selected individuals and their parents were sequenced with RADseq technology in 2013 and 2016, and each had PE reads data of > 2× genome coverage [50]. The accession number prefixed with SRR of each individual data was listed in Appendix A and all the data reads are available in the NCBI SRA database (http://www.ncbi.nlm.nih.gov/Traces/sra, accessed on 1 January 2022).

### 4.2. SNP Genotyping

First of all, the PE reads of each individual were filtered to generate high-quality (HQ) data using the NGS QC toolkit with default parameters [51]. Then, we mapped the HQ reads to the improved genome sequence of the male parent *P. simonii* [50] to call SNP genotypes of each individual using the software BWA [70], SAMtools and BCFtools [71]. The calling procedures were as follows: (1) mapping each individual HQ reads to the reference genome to obtain a sequence alignment/map (SAM) format file using the mem command of BWA with default parameters; (2) converting each SAM file to BAM format and sorting and indexing it with SAMtools; (3) producing a BCF file with each sorted BAM file using the command *bcftools mpileup -Obuzv -a AD,INFO/AD –f*; (4) generating VCF files with the command *bcftools call -m -v -f gq* for each parent; (5) filtering SNPs from the two parental VCF files such that the allele DP (read coverage depth) ≥ 3 and the GQ (genotyping quality) > 30, and merging the two parental SNP sites into a list site file; (6) creating VCF files for all individuals with the command *bcftools call -m -v -f gq* using the list site file generated above; (7) extracting all SNP genotypes of each individual from its VCF file, satisfying that the allele DP ≥ 3 and GQ > 30 for a heterozygous genotype and that the allele DP ≥ 5 for a homozygous genotype; and (8) merging all individual SNP genotypes into one SNP dataset.

As indicated in the previous studies [43,46,50], the SNPs were mainly segregated in the patterns of *aa* × *ab* and *ab* × *aa*, where the first two letters represent the female parental genotype and the last two the male parental genotype. We extracted two SNP datasets, one for the segregation type of *aa* × *ab* and the other for the type of *ab* × *aa*. Meanwhile, we filtered out those SNPs that were seriously deviated from the Mendelian segregation ratio of 1:1 (*p* > 0.01). Moreover, those SNPs with more than 20% missing genotypes across the progeny were also removed from the dataset.

We further selected RAD tags that contained two SNPs, one with a segregation pattern of *ab* × *aa* and the other *aa* × *ab* (Figure 1A), to investigate the difference of CO patterns between the two parents. The genotypes of the adjacent RAD tags in the F1 progeny allowed us to estimate the female and male parental CO rates (Figure 1B,C), and even to compare the CO interference between the two species (Figure 5).

### 4.3. Comparison of CO Rates between Two Parents

For any two adjacent RAD tags as described above (Figure 1A), assume the female *P. deltoides* and the male *P. simonii* CO rates to be $r_{f}$ and $r_{m}$, respectively. In the F1 hybrid population, if we denote the counts of the four genotypes as $n_{11}$,$n_{12}$,$n_{13}$, and $n_{14}$ at the two SNPs that are contained in the two RAD tags and segregate in *ab* × *aa* (Figure 1B), and the counts of the four genotypes as $n_{21}$,$n_{22}$,$n_{23}$, and $n_{24}$ at the two SNPs segregating in *aa* × *ab* (Figure 1C), then the likelihood of $r_{f}$ and $r_{m}$ can be expressed, under the assumption of coupling linkage phase, as
(1)$$L_{1}(r_{f},\;r_{m})=\frac{n_{1}!}{n_{11}!n_{12}!n_{13}!n_{14}!}{\left[\frac{1}{2}\left(1-r_{f}\right)\right]}^{n_{11}+n_{14}}{\left(\frac{1}{2}r_{f}\right)}^{n_{12}+n_{13}}\times \frac{n_{2}!}{n_{21}!n_{22}!n_{23}!n_{24}!}{\left[\frac{1}{2}\left(1-r_{m}\right)\right]}^{n_{21}+n_{24}}{\left(\frac{1}{2}r_{m}\right)}^{n_{22}+n_{23}}$$
where $n_{1}=n_{11}+n_{12}+n_{13}+n_{14}$ and $n_{2}=n_{21}+n_{22}+n_{23}+n_{24}$. The maximum likelihood estimates (MLEs) of $r_{f}$ and $r_{m}$ can be easily obtained as
(2)$${\hat{r}}_{f}=\frac{n_{12}+n_{13}}{n_{1}}\;\;\;\;\mathrm{and}\;\;\;\;r_{m}=\frac{n_{22}+n_{23}}{n_{2}}$$

To test if the CO rates of the two sexes are equal or not, the null hypothesis is defined as $H_{10}:\;\;\;r_{f}=r_{m}=r$, under which the likelihood can be written as
(3)$$L_{10}(r)=\frac{n_{1}!}{n_{11}!n_{12}!n_{13}!n_{14}!}\times \frac{n_{2}!}{n_{21}!n_{22}!n_{23}!n_{24}!}{\left[\frac{1}{2}\left(1-r\right)\right]}^{n_{11}+n_{14+}n_{21}+n_{24}}{\left(\frac{1}{2}r\right)}^{n_{12}+n_{13}+n_{22}+n_{23}}$$
and the ML estimate is derived as
(4)$$\hat{r}=\frac{n_{12}+n_{13}+n_{22}+n_{23}}{n_{1}+n_{2}}$$

Therefore, the log-likelihood ratio (LR) of the full model over the null model is expressed as
(5)$$LR_{1}=2\log\left[\frac{L_{1}({\hat{r}}_{f},\;{\hat{r}}_{m})}{L_{10}(\hat{r})}\right]$$
which is asymptotically distributed as a chi-square distribution with 1 degree of freedom.

### 4.4. Analysis of CO Interference in Both Parents

Suppose that three adjacent RAD tags 1, 2 and 3 are located in such an order on thw chromosome, each containing two SNPs with different segregation types (Figure 5A). For the SNPs of 1, 3, and 5 that segregate in *ab* × *aa* on the three RAD tags (Figure 5B), let $C_{f}$ denote the CoC between the two adjacent SNP intervals for the female parent and $r_{fij}$ the frequencies of crossovers occurred (i=1) or not (i=0) in the first interval, and occurred (j=1) or not (j=0) in the second interval. Then, we have the relationships between the CoC and the CO rates under the assumption of coupling linkage phases, as the following:(6)$$\left\{ \begin{array}{l} r_{f11}=r_{f1}r_{f2}C_{f}\\ r_{f01}=r_{f2}-r_{f1}r_{f2}C_{f}\\ r_{f10}=r_{f1}-r_{f1}r_{f2}C_{f}\\ r_{f00}=1-r_{f1}-r_{f2}+r_{f1}r_{f2}C_{f} \end{array}\right.$$
where $r_{f1}$ and $r_{f2}$ are the CO rates of the first and second intervals, respectively. Therefore, the likelihood of the CO rates and the CoC can be written as
(7)$$L_{f}(C_{f})=\frac{n_{1}!}{\prod_{j=1}^{8}n_{1j}!}{\left(\frac{1}{2}r_{f00}\right)}^{n_{11}+n_{18}}{\left(\frac{1}{2}r_{f01}\right)}^{n_{12}+n_{17}}{\left(\frac{1}{2}r_{f10}\right)}^{n_{14}+n_{15}}{\left(\frac{1}{2}r_{f11}\right)}^{n_{13}+n_{16}}$$
from which the MLEs of the parameters $r_{f1}$, $r_{f2}$, and $C_{f}$ can be obtained as the following (Appendix A):(8)$$\left\{ \begin{array}{l} {\hat{r}}_{f1}=\frac{1}{n_{1}}\left(n_{13}+n_{14}+n_{15}+n_{16}\right)\\ {\hat{r}}_{f2}=\frac{1}{n_{1}}\left(n_{12}+n_{13}+n_{16}+n_{17}\right)\\ {\hat{C}}_{f}=\frac{n_{13}+n_{16}}{n_{1}{\hat{r}}_{f1}{\hat{r}}_{f2}} \end{array}\right.$$
where $n_{f}=n_{11}+n_{12}+\ldots +n_{18}$. To test if the crossover interference exists between the two adjacent SNP intervals, we applied the LR statistic as
(9)$$LR_{f}=2\log\left[\frac{L_{f}({\hat{C}}_{f})}{L_{f}(C_{f}=1)}\right]$$
where $n_{2j}$ and $r_{mij}$ denote the counts and frequencies of crossovers similar to Equation (6) containing the CO rates of the first and second SNP intervals and the parameter $C_{m}$. Meanwhile, the LR statistic for testing the existence of the CO interference can be applied and has the same form as Equation (9).

The above analyses of CO interference are under the assumption of the coupling linkage phase, i.e., the Case 1 of linkage phases as described in Appendix A. For the other cases of linkage phases, the corresponding likelihoods and LR statistics largely have the same forms as above, but with different forms of exponents in Equations (7) and (10), which lead to the MLEs of the parameters as modified versions of Equation (8).

To compare the difference of CO interference between the two parents, we used the R package xoi to estimate the strength of interference by fitting a gamma model to the distribution of observed crossover [53]. The parameter ν of the fitted gamma distribution is a measure of the strength of CO interference, corresponding to no, positive, and negative interference for ν = 1, ν>1, and ν<1, respectively. The comparison of interference strength between two sexes was performed using a student’s *t* test on the values estimated from each chromosome.

## Figures and Tables

**Figure 1 plants-11-01046-f001:**
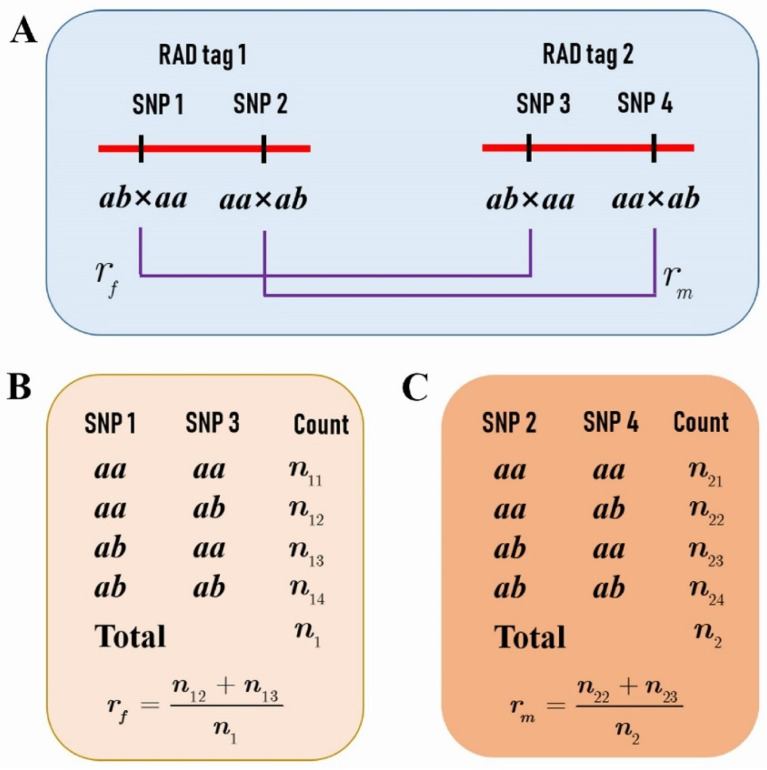
The *P. deltoides* and *P. simonii* crossover (CO) rates between two restriction site-associated DNA (RAD) tags. (**A**) Each RAD tag contains two SNPs, one with a segregation pattern of *ab* × *aa* and the other *aa* × *ab*. (**B**) The counts of genotypes at single nucleotide polymorphisms (SNPs) 1 and 3 in the progeny provide the estimate of the female parental *P. deltoides* CO rate assuming the linkage phase in coupling. (**C**) The counts of genotypes at SNPs 2 and 4 provide the estimate of the male parental *P. simonii* CO rate assuming the linkage phase in coupling.

**Figure 2 plants-11-01046-f002:**
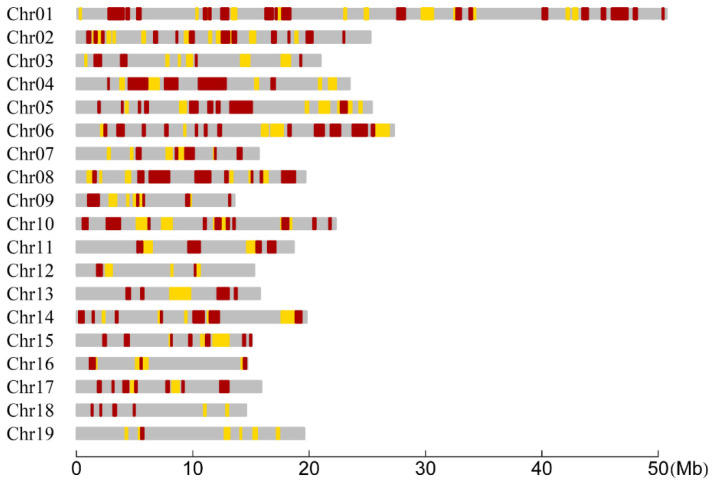
Distribution of RAD tag intervals and regions within chromosomes for significantly differential CO rates between the *P. deltoides* and *P. simonii*. The yellow and red bars represent the intervals when the significances are at the 0.05 and 0.01 levels, respectively. Two or more successive significant intervals constitute a significant region.

**Figure 3 plants-11-01046-f003:**
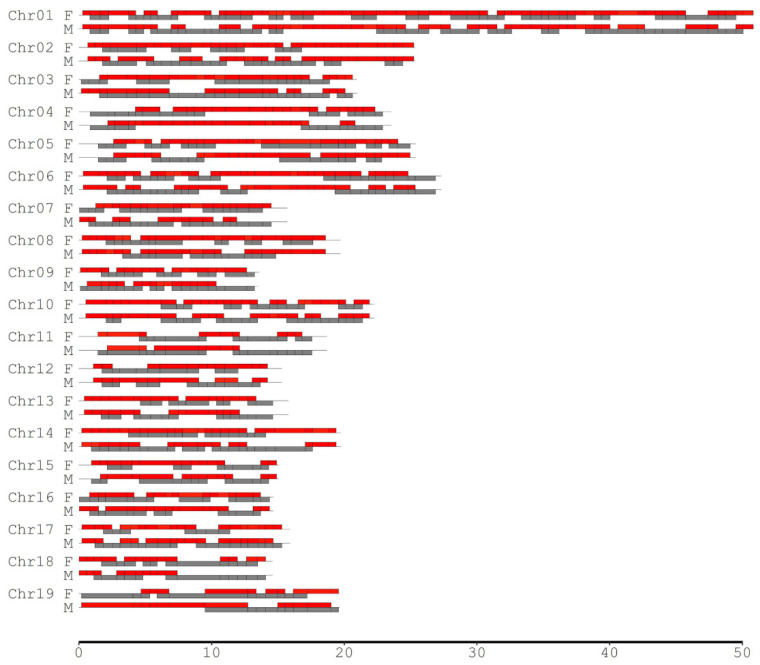
Distribution of the adjacent intervals where CO interference significantly existed across chromosomes in the male (M) *P. simonii* and female (F) *P. deltoides*. The minimum interval length was set to be 0.5 Mb. The red and gray bars indicate the regions with and without CO inference, respectively.

**Figure 4 plants-11-01046-f004:**
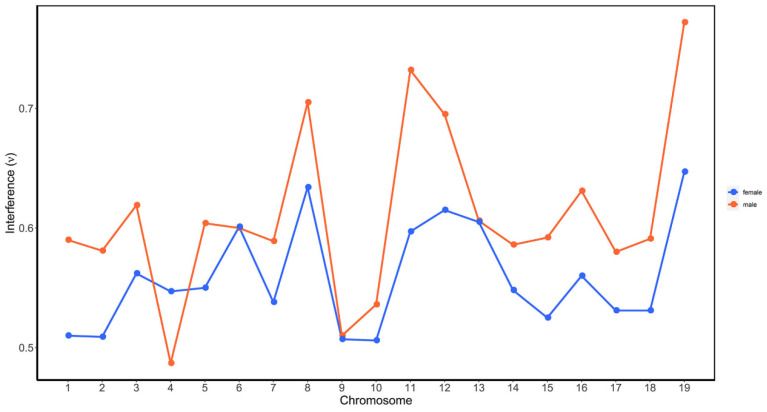
CO interference parameters of the female *P. deltoides* (red) and male *P. simonii* (blue across the 19 chromosomes. The interference parameter ν was estimated from the gamma model as described in Broman and Weber [53].

**Figure 5 plants-11-01046-f005:**
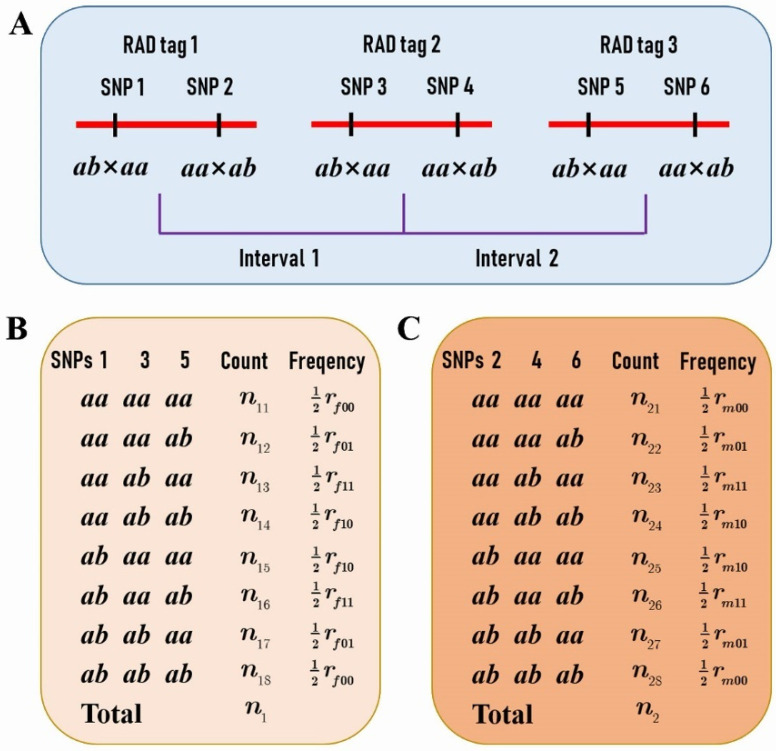
Three RAD tags provide information on CO interference for the female *P. deltoides* and male *P. simonii*. (**A**) Three RAD tags constitute two adjacent marker intervals. Each RAD tag contains two SNPs, one with a segregation pattern of *ab* × *aa* and the other *aa* × *ab*. (**B**) The counts and frequencies of genotypes at SNPs 1, 3, and 5 in the progeny provide the estimate of the CoC of the female *P. deltoides* between the two intervals, assuming the linkage phases in coupling. (**C**) The counts and frequencies of genotypes at SNPs 2, 4, and 6 in the progeny provide the estimate of the CoC of the male *P. simonii* between the two intervals assuming the linkage phases in coupling.

**Table 1 plants-11-01046-t001:** Summary of adjacent RAD tag interval lengths and CO rates within chromosomes in the female parent *Populus deltoides* and the male parent *Populus simonii*.

Chrom.	RAD Tag Number	Length Range (Kb)	Length Mean (Kb)	Female CO Rate Range	Female CO Rate Mean	Male CO Rate Range	Male CO Rate Mean
Chr01	151	101~1492	337	0.005~0.150	0.053	0.004~0.166	0.040
Chr02	88	102~1191	282	0.008~0.111	0.047	0.005~0.118	0.043
Chr03	67	100~1502	313	0.005~0.113	0.046	0.004~0.117	0.044
Chr04	67	103~1370	338	0.004~0.137	0.044	0.008~0.111	0.040
Chr05	71	105~2195	336	0.008~0.202	0.060	0.004~0.100	0.040
Chr06	83	100~1490	324	0.008~0.192	0.057	0.005~0.202	0.046
Chr07	49	102~1175	301	0.004~0.170	0.052	0.009~0.102	0.042
Chr08	51	110~1566	374	0.004~0.165	0.056	0.014~0.225	0.066
Chr09	54	101~1310	248	0.008~0.147	0.043	0.008~0.095	0.038
Chr10	82	100~1056	265	0.005~0.164	0.050	0.004~0.123	0.039
Chr11	41	108~2405	403	0.004~0.127	0.060	0.009~0.123	0.039
Chr12	37	103~2025	365	0.009~0.145	0.050	0.008~0.159	0.052
Chr13	42	100~1755	346	0.008~0.152	0.048	0.004~0.089	0.039
Chr14	60	106~1990	325	0.004~0.203	0.045	0.005~0.147	0.045
Chr15	46	102~2311	314	0.012~0.110	0.044	0.004~0.136	0.040
Chr16	40	109~1659	374	0.005~0.167	0.052	0.005~0.158	0.043
Chr17	49	102~869	314	0.008~0.148	0.053	0.009~0.101	0.042
Chr18	47	100~2837	306	0.004~0.201	0.047	0.004~0.204	0.044
Chr19	32	104~3687	625	0.008~0.120	0.056	0.012~0.158	0.054

**Table 2 plants-11-01046-t002:** Distribution of the numbers of significant SNP intervals and regions over different lengths within chromosomes. The CO rates of these intervals are significantly different in the two parents. The significant region is constituted of two or more continuous significant intervals.

Chrom.	Interval	Sig. Interval	Sig. Region	100–500 kb	500 Kb–1 Mb	≥1 Mb	Cover. (Mb) ^a^	Percent (%) ^b^	Chrom. Size (Mb) ^c^
Chr01	150	41	11	15	6	4	13.06	25.69	50.84
Chr02	87	32	8	13	3	1	6.86	27.09	25.34
Chr03	66	13	2	8	3	0	3.70	17.63	20.97
Chr04	66	16	4	5	2	3	7.92	33.65	23.53
Chr05	70	23	5	9	3	2	7.32	28.83	25.39
Chr06	82	25	8	9	4	3	8.46	30.98	27.31
Chr07	48	10	2	6	0	1	3.43	21.82	15.71
Chr08	50	20	7	5	3	3	7.43	37.72	19.71
Chr09	53	11	3	3	3	0	3.28	24.14	13.60
Chr10	81	24	4	9	1	3	6.35	28.53	22.25
Chr11	40	15	4	3	1	2	3.94	21.04	18.70
Chr12	36	7	3	3	1	0	1.65	10.75	15.30
Chr13	41	7	2	3	0	2	3.58	22.65	15.80
Chr14	59	17	5	6	2	1	5.21	26.35	19.76
Chr15	45	12	3	8	0	1	3.43	22.75	15.07
Chr16	39	8	3	1	1	1	2.13	14.52	14.66
Chr17	48	13	2	5	3	0	3.23	20.29	15.92
Chr18	46	6	0	6	0	0	1.01	6.91	14.60
Chr19	31	7	1	6	0	0	1.81	9.20	19.64
Total	1138	307	77	123	36	27	93.78	23.80	394.11

^a^ The total coverage length in Mb of the significant intervals in the chromosome. ^b^ The percentage of the total coverage of significant intervals over the chromosome length. ^c^ The chromosome size of *Populus simonii* [50].

**Table 3 plants-11-01046-t003:** The number of pairs of adjacent interference-occurred intervals with their coverage in bracket across different minimum interval lengths in the genomes of the female (F) *P. deltoides* and the male (M) *P. simonii*.

Parent	CoC	0.5 Mb	1.0 Mb	2.0 Mb	3.0 Mb	4.0 Mb	5.0 Mb
F	>1	214	64	13	9	3	0
	<1	76	34	29	25	25	14
	Total (Coverage)	290 (79.86%)	98 (57.05%)	42 (54.79%)	34 (46.17%)	28 (55.35%)	14 (35.10%)
M	>1	138	37	8	2	1	0
	<1	95	40	31	34	18	9
	Total (Coverage)	233 (69.77%)	77 (46.35%)	39 (52.44%)	36 (55.50%)	19 (38.28%)	9 (22.78%)
Both F and M	>1	81	18	1	1	0	0
	<1	17	3	12	14	11	4
	Total (Coverage)	168 (52.61%)	37 (24.21%)	16 (23.81%)	17 (27.46%)	12 (24.26%)	4 (9.20%)

## Data Availability

The RADseq data for all individuals are available in the NCBI SRA database (http://www.ncbi.nlm.nih.gov/Traces/sra, accessed on 1 January 2022) with accession numbers listed in Appendix A.

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
