# Peer review of "A Novel Strategy to Reveal the Landscape of Crossovers in an F1 Hybrid Population of *Populus deltoides* and *Populus simonii"

_plants, 2022, doi:10.3390/plants11081046_

Round 1

Reviewer 1 Report

This interesting study on the frequency and chromosomal distribution of crossovers in the progenies of a cross between two poplars provides detailed and new information on a topic which has received little attention in poplars and other tree species in the past. It is also of general interest for plant geneticists since, as correctly pointed out by the authors, the method described here can be used for other outcrossing plant species as well.

The following two aspects might be considered in a revised version of the manuscript: (i) Results refer to progenies from a single cross with one female  Populus deltoides tree and one male P. simonii tree. It should be pointed out more clearly in the discussion (e.g. line 256ff.), that results refer to this specific cross only, but not to the two species "in general". This might be done by expanding the recommendations for further studies (line 269ff.) and a few more clarifications throughout the text (e.g. line 86ff.; but see also minor comment below). (ii) Potential applications of the results might be concisely, but in more detail discussed in a breeding context, for example with reference to QTL studies.

Minor comments: (i) Main results are reported at the end of the introduction (line 86ff.). This is not necessary in this paragraph since it is a repetition and somehow confusing for the reader. It should be deleted or rephrased. (ii) Check spelling of P. deltoides (e.g. line 12, line 269) throughout the text (not P. deltoids).       

Author Response

Response to Reviewer 1

This interesting study on the frequency and chromosomal distribution of crossovers in the progenies of a cross between two poplars provides detailed and new information on a topic which has received little attention in poplars and other tree species in the past. It is also of general interest for plant geneticists since, as correctly pointed out by the authors, the method described here can be used for other outcrossing plant species as well.

The following two aspects might be considered in a revised version of the manuscript:

(i) Results refer to progenies from a single cross with one female  Populus deltoides tree and one male P. simonii tree. It should be pointed out more clearly in the discussion (e.g. line 256ff.), that results refer to this specific cross only, but not to the two species "in general". This might be done by expanding the recommendations for further studies (line 269ff.) and a few more clarifications throughout the text (e.g. line 86ff.; but see also minor comment below).

RE: Thank you very much for your constructive comments.  We modified the sentence in Line 269 as follows: “We therefore revealed the genome-wide CO patterns between the two parents in the F1 population, though the results may not be suitable for all species in Populus”.

(ii) Potential applications of the results might be concisely, but in more detail discussed in a breeding context, for example with reference to QTL studies.

RE: Thank you very much for this suggestion. We added several sentences in the end of the last paragraph in Discussion to detail the effect of CO interference on QTL mapping, as follows:

“It is worth to note that the CO interference affects QTL studies in two aspects. On the one hand, genetic linkage maps that incorporated CO interference could improve the power of detecting QTLs. On the other hand, the popular QTL statistical methods such as interval mapping [68] and composite interval mapping [69] were established under the assumption of without CO interference. Without any doubt, if CO interference was incorporated into the QTL mapping models, it would greatly improve their ability of identifying QTLs.”

Minor comments:

(i) Main results are reported at the end of the introduction (line 86ff.). This is not necessary in this paragraph since it is a repetition and somehow confusing for the reader. It should be deleted or rephrased.

RE: Thanks for this constructive suggestion. We deleted the sentence and made some modification thereafter.

(ii) Check spelling of P. deltoides (e.g. line 12, line 269) throughout the text (not P. deltoids).    

RE: Thank you very much for pointing out this spelling error. We checked carefully throughout the manuscript and corrected the error in the two places.

Reviewer 2 Report

The manuscript presented impressed me by its content. It is original in studying overall patterns of crossing over for Eucaryotes in forest model species of Populus. The methods presented are not only adequate but novel, and their methods can be applied to other species for which F1 populations are available, so I anticipate a good amount of citations of this paper regarding its methodology. I could not find any flaw in the rationale and execution of the analyses, and the results are quite interesting. 

The paper is also well-writen, and with an excellent English language standard, and I belive it can be easily published with little revision. 

I recommend the paper for its quality, broad interest to readers in plant genetics, and methodological novelty.  

Author Response

Response to Reviewer 2

The manuscript presented impressed me by its content. It is original in studying overall patterns of crossing over for Eucaryotes in forest model species of Populus. The methods presented are not only adequate but novel, and their methods can be applied to other species for which F1 populations are available, so I anticipate a good amount of citations of this paper regarding its methodology. I could not find any flaw in the rationale and execution of the analyses, and the results are quite interesting.

The paper is also well-writen, and with an excellent English language standard, and I belive it can be easily published with little revision.

I recommend the paper for its quality, broad interest to readers in plant genetics, and methodological novelty. 

RE: Thank you very much for your highly positive comments and recommendation.
